# Essential yet limited role for CCR2+ inflammatory monocytes during *Mycobacterium tuberculosis*-specific T cell priming

Miriam Samstein[1,2,3]*, Heidi A Schreiber[4], Ingrid M Leiner[1], Bože Sušac[1], Michael S Glickman[1,2], Eric G Pamer[1,2]*

[1]Program in Immunology and Microbial Pathogenesis, Weill Cornell Graduate School of Medical Sciences, New York, United States; [2]Infectious Disease Service, Department of Medicine, Memorial Sloan Kettering Cancer Center, New York, United States; [3]Weill Cornell, Rockefeller, Sloan-Kettering Tri-Institutional MD-PhD Program, New York, United States; [4]Laboratory of Molecular Immunology, The Rockefeller University, New York, United States

**Abstract** Defense against infection by *Mycobacterium tuberculosis* (Mtb) is mediated by CD4 T cells. CCR2+ inflammatory monocytes (IMs) have been implicated in Mtb-specific CD4 T cell responses but their in vivo contribution remains unresolved. Herein, we show that transient ablation of IMs during infection prevents Mtb delivery to pulmonary lymph nodes, reducing CD4 T cell responses. Transfer of MHC class II-expressing IMs to MHC class II-deficient, monocyte-depleted recipients, while restoring Mtb transport to mLNs, does not enable Mtb-specific CD4 T cell priming. On the other hand, transfer of MHC class II-deficient IMs corrects CD4 T cell priming in monocyte-depleted, MHC class II-expressing mice. Specific depletion of classical DCs does not reduce Mtb delivery to pulmonary lymph nodes but markedly reduces CD4 T cell priming. Thus, although IMs acquire characteristics of DCs while delivering Mtb to lymph nodes, cDCs but not moDCs induce proliferation of Mtb-specific CD4 T cells.

*For correspondence: mb. eichenbaum@gmail.com (MS); pamere@mskcc.org (EGP)

**Competing interests:** The authors declare that no competing interests exist.

**Reviewing editor**: Antonio Lanzavecchia, Institute for Research in Biomedicine, Switzerland

## Introduction

Inflammatory monocytes (IMs) express Ly6c, CD11b and CCR2, a chemokine receptor that facilitates emigration of IMs from the bone marrow (*Serbina and Pamer, 2006*). Although IMs make important contributions to innate immune defense during infection, recent studies also implicate IMs in priming of CD4 T cell responses during fungal, viral and parasitic infections (*Traynor et al., 2000*; *Leon et al., 2007*; *Edismo et al., 2009*; *Hohl et al., 2009*; *Rivera et al., 2011*). Experiments using CCR2-deficient mice also suggested that monocytes contribute to T-cell-mediated defense against Mtb. IMs, however, can also serve as permissive host cells for Mtb in later stages of infection, suggesting that IMs can both restrict and enhance Mtb infection (*Antonelli et al., 2010*). Because recruitment of IMs in CCR2-deficient mice is defective throughout early, intermediate and late stages of infection, it has not been possible to specifically define the role of monocytes at different times during infection (*Peters et al., 2001*; *Scott and Flynn, 2002*; *Peters et al., 2004*). Therefore, we used CCR2-DTR mice to transiently deplete IMs and other CCR2-expressing cells during discrete stages of Mtb infection.

## Results and discussion

Administration of DT to these mice depletes inflammatory monocytes from the lung and mLN (*Figure 1—figure supplement 1*). CCR2-DTR mice that received three doses of DT surrounding the

**eLife digest** Tuberculosis is a disease that kills more than one million people every year. It is caused by mycobacteria, notably *Mycobacterium tuberculosis*, and the World Health Organization estimates that about one third of the world's population has latent tuberculosis, although only one person in 10 goes on to develop an active infection. Understanding why some individuals develop active infections, whereas most do not, could help with the development of a vaccine to prevent tuberculosis and/or new treatments for the disease. Disappointing results from vaccine trials and the emergence of drug-resistant strains of tuberculosis have increased the need for more research into the interactions between mycobacteria and the human immune system.

Tuberculosis is spread when an infected person coughs or sneezes and someone else inhales the mycobacteria spread by the first person. When *M. tuberculosis* first enters the human respiratory tract, the innate immune system tries to identify and destroy cells that have been infected. However, if this initial response is not effective, the *M. tuberculosis* can persist in the lungs and trigger the adaptive immune response. This involves CD4 T cells working to eliminate the infection, but our understanding of the adaptive immune response is not complete.

Samstein et al. probed the role that immune cells known as inflammatory monocytes play in the adaptive immune response. Previous research has suggested that inflammatory monocytes may develop into dendritic cells that directly prime the CD4 T cells to respond when the lung has been infected. However, Samstein et al. demonstrate that the inflammatory monocytes carry *M. tuberculosis* from the lungs of infected mice to the draining lymph nodes during the second week of infection. These monocytes develop many of the characteristics of dendritic cells, but they do not activate the CD4 T cells.

Samstein et al. show that dendritic cells, contrary to previous evidence, are not necessary for the transport of the *M. tuberculosis* from the lungs to the draining lymph nodes. Without the dendritic cells, however, fewer CD4 T cell are primed in the lymph nodes. Samstein et al. suggest that the inflammatory monocytes play a crucial role by transporting the live bacteria to the lymph nodes. And once in the lymph nodes, the monocytes transfer invading antigens to dendritic cells to initiate the production of the CD4 T cells to lead the fight against the infection.

time of aerosol infection did not differ from control mice in terms of mycobacterial growth in the lungs (*Figure 1A*), indicating that CCR2-expressing cells are not required for the establishment of pulmonary infection following Mtb inhalation. Previous studies have demonstrated that CD4 T-cell priming occurs approximately 7–10 days following inhalational challenge (*Wolf et al., 2007*; *Gallegos et al., 2008*). Depletion of CCR2-expressing cells 7, 9 and 11 days following infection resulted in a threefold increase in the number of live Mtb in the lungs on day 15 (*Figure 1B*). To further investigate this finding, we depleted monocytes 7–11 days following infection and quantitatively cultured Mtb from lungs and lymph nodes 12 days following infection. Although monocyte-depleted and control mice had similar Mtb CFUs in lungs at this time (*Figure 1C*), Mtb CFUs were markedly reduced in mLNs of mice treated with DT (1D).

We reasoned that reducing live *M. tuberculosis* transport from the lung to the draining lymph node following monocyte depletion on days 7–11 post infection might delay priming of Mtb-specific CD4 T cells. To address this, we adoptively transferred naïve, Thy1.1-marked, ESAT-6-specific CD4 T cells (C7 T cells) into CCR2-DTR mice prior to aerosol Mtb infection (*Gallegos et al., 2008*). Monocytes were depleted 7–11 days following infection, and mediastinal lymph nodes and lungs were harvested 12 and 21 days following infection. In comparison to mLNs obtained from PBS treated mice, mLNs obtained from monocyte-depleted mice were smaller, with significantly reduced total and CD4 T cell numbers (*Figure 2A*). C7 T cells had proliferated in mLNs of control mice 12 days after infection but had not expanded in monocyte-depleted mice (*Figure 2B,D*). Reduced C7 T cell proliferation in mLNs resulted in reduced C7 T cell frequencies in lungs of monocyte-depleted mice 21 days following infection (*Figure 2C,D*). Transfer of naïve C7 T cells to control mice resulted in a four to fivefold reduction of Mtb CFUs in lungs, suggesting that supplementing the endogenous T cell repertoire with additional naive, ESAT-6-specific CD4 T cells enhances immune defense against Mtb infection (*Figure 2E*). Depletion of monocytes abrogated enhanced protection resulting from addition of naive C7 T cells (*Figure 2E*).

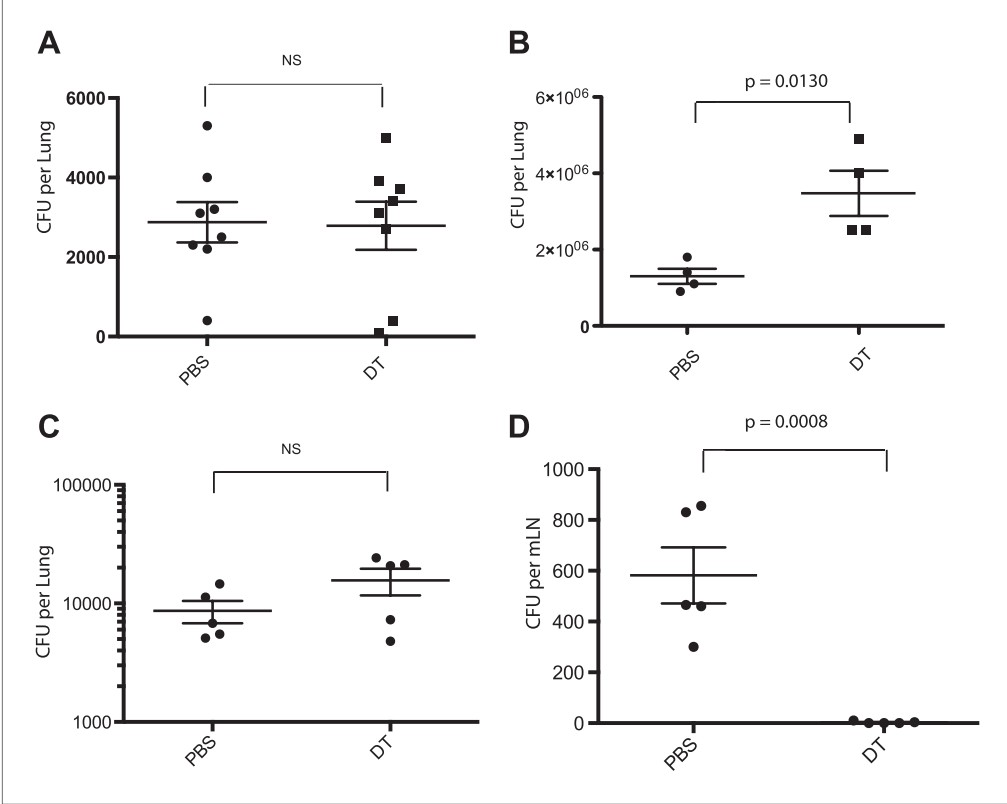

**Figure 1**. Depletion of inflammatory monocytes during the second week of *M. tuberculosis* infection abrogates transport of live bacteria to mLNs and increases pulmonary bacterial burden. (**A**) CFU plots from the lungs of CCR2-DTR mice receiving DT or PBS on days −1, 0 and 1 and harvested on day 7. (**B**) CFU plots of the lungs of CCR2-DTR mice receiving DT or PBS on days 7, 9 and 11 and harvested on day 15 post infection. CFU counts from the lungs (**C**) and mLN (**D**) of CCR2-DTR mice given DT treatment on days 7, 9 and 11 and harvested on day 12. Each dot represents an individual mouse. Error bars denote SEM. Data are representative of two independent experiments.

The following figure supplements are available for figure 1:

**Figure supplement 1**. DT administration depletes IMs from the lungs and mLNs of infected mice.

Although IMs represent the most prevalent CCR2-expressing cell population, subsets of NK cells, dendritic cells and CD4 T cells can also express CCR2. To exclude the possibility that depletion of a non-monocyte cell population resulted in loss of Mtb transport from lungs to mLNs, we purified IMs from CCR2-GFP mice by sorting GFP+ cells that did not express NK1.1, CD4, FLT3 and C-kit. This strategy did not require antibody staining for trafficking molecules (CCR2 and CD11b) and eliminated NK cells, CD4 T cells, dendritic cell progenitors and hematopoietic stem cells, yielding a highly purified (>99% pure) population of IMs (*Figure 3—figure supplement 1*). We adoptively transferred $2 \times 10^6$ IMs on day 8 and 10 of infection while depleting CCR2+ cells on days 7, 9 and 11. The transferred IMs were detectable in lungs and mLNs and down-regulated expression of CCR2 and CD11b and up-regulated expression of MHC class II, CD11c and CD103 during trafficking from the lung to mLNs (*Figure 3A,B*). Administration of IMs to monocyte-depleted CCR2-DTR mice enhanced live Mtb transport to mLNs, as detected 12 days following infection (*Figure 3C*). Infusion of IMs to WT mice did not enhance delivery of Mtb to mLNs, suggesting that IMs are not limiting during the first 2 weeks of Mtb infection. Adoptive transfer of IMs to monocyte-depleted CCR2-DTR recipient mice rescued priming of C7 T cells (*Figure 3D,E*).

While the CCR2 chemokine receptor is required for egress of IMs from bone marrow (*Serbina and Pamer, 2006*), the role of CCR2 in trafficking of IMs into infected lungs or from the sites of lung infection to draining lymph nodes remains incompletely resolved. To determine whether CCR2 is

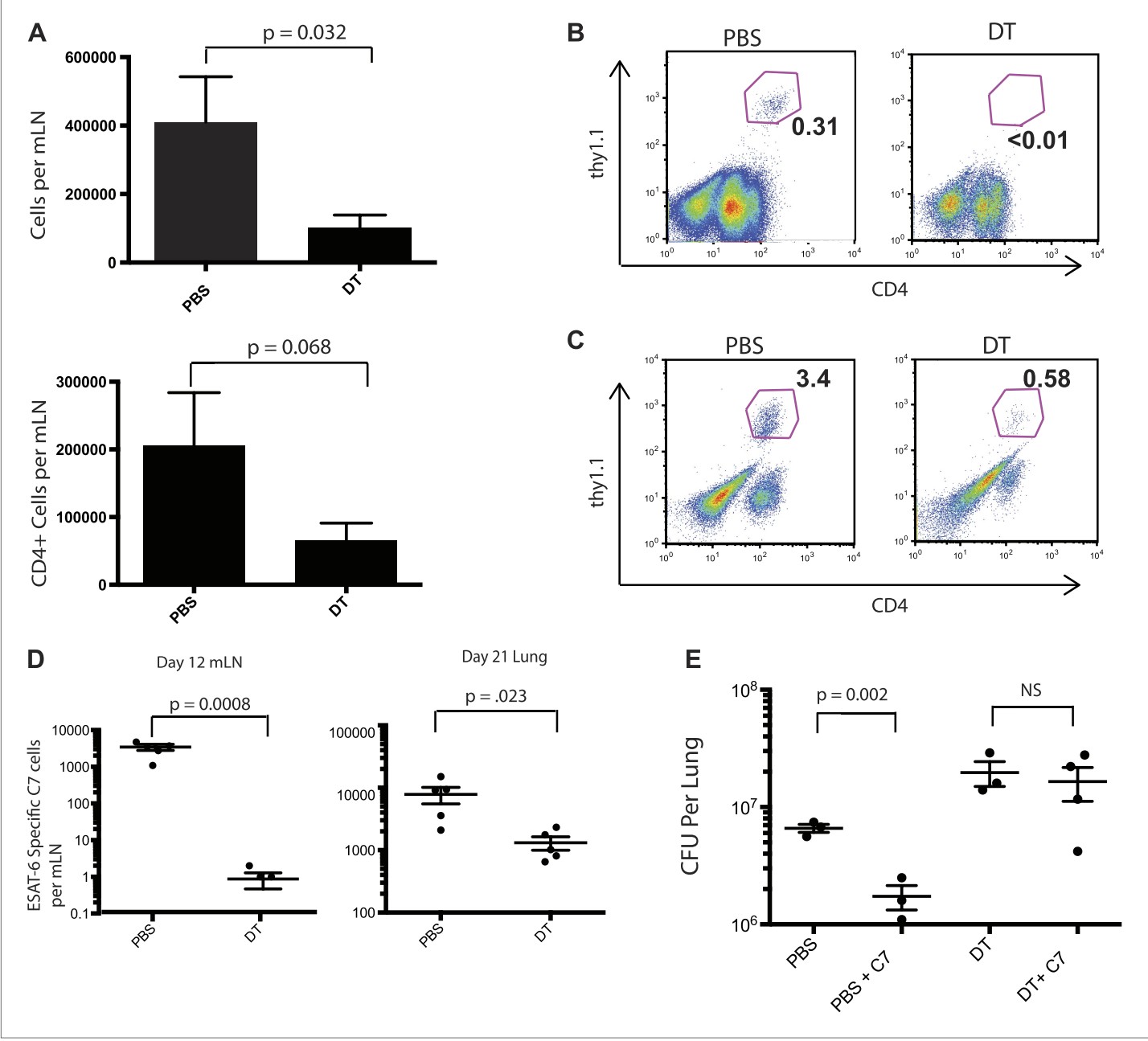

**Figure 2**. Priming of ESAT-6-specific CD4 T cells is reduced in CCR2-DTR mice depleted during the second week of infection. (**A**) Total cell counts and CD4 T cell counts in mLNs of mice on day 12 post infection after treatment with DT or PBS on days 7, 9 and 11. CCR2-DTR mice received naive ESAT-6-specific C7 T cells the day before infection and were treated with DT or PBS on days 7, 9 and 11, mLNs (**B**) were harvested on day 12, lungs (**C**) were harvested on day 21 and ESAT-6-specific C7 T cells were visualized. (**D**) Cumulative data from experiments shown in (**B**) and (**C**) showing the total number of ESAT-6-specific C7 T cells in mLNs and lungs. (**E**) CFU plots of lungs of mice that received naive EAST-6-specific C7 T cells and were treated with DT or PBS, as indicated, and were harvested on day 21 post infection. Each dot represents an individual mouse. Five mice per group are included in the bar graphs shown in (**A**). Error bars denote SEM. Data are representative of three independent experiments.

required for trafficking to lungs and mLNs during Mtb infection, we purified IMs from CCR2-deficient, CCR2-GFP mice for adoptive transfer into Mtb-infected, CCR2-DTR recipient mice that were monocyte depleted between 7 and 11 days following infection. Quantitative culture 12 days following Mtb infection demonstrated that wild-type and CCR2-deficient IMs were equivalent at delivering Mtb to mLNs, indicating that CCR2-signaling does not contribute to trafficking of IMs between these sites (*Figure 3F*).

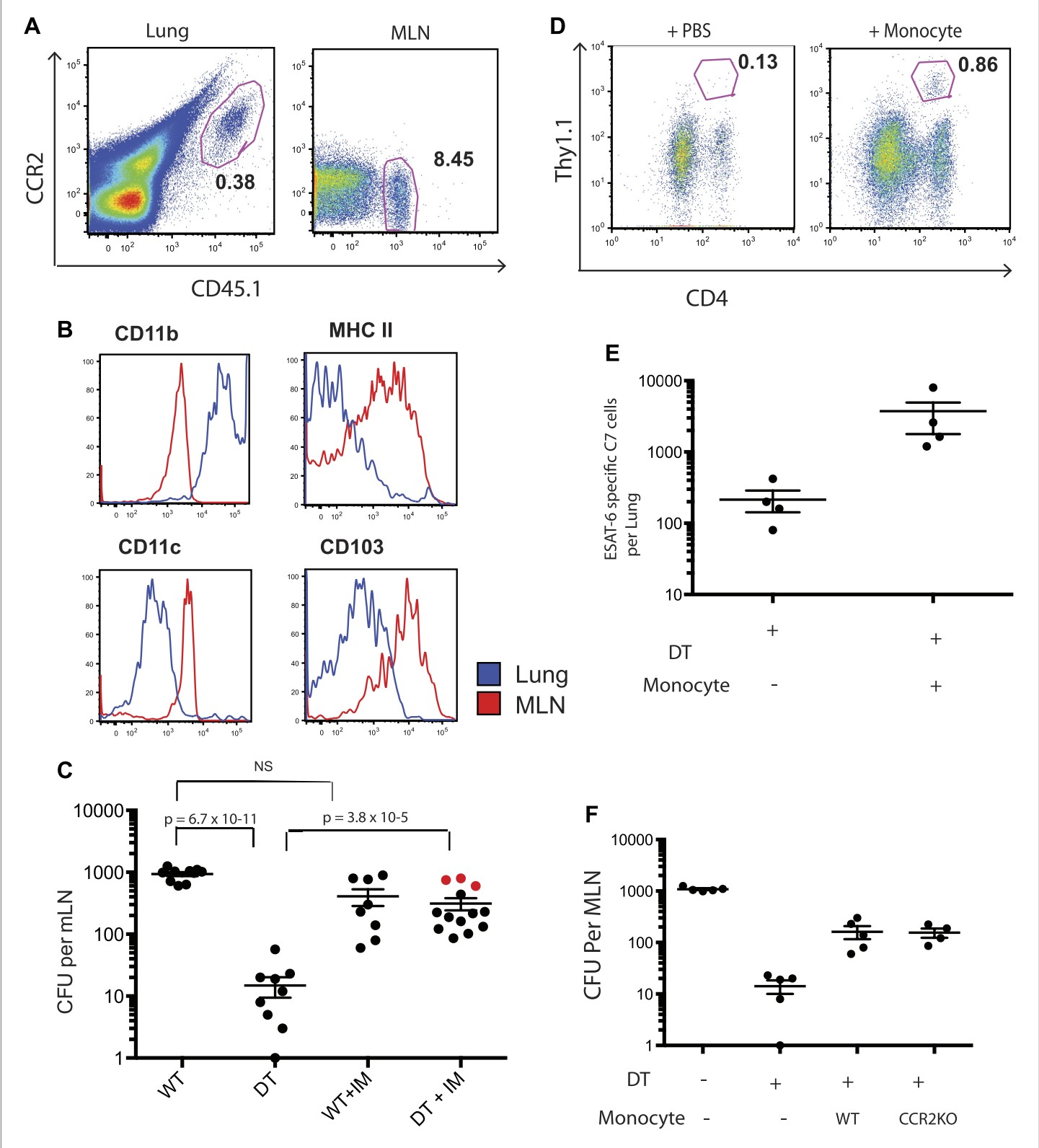

**Figure 3**. Adoptive transfer of highly purified inflammatory monocytes can rescue antigen transport and CD4 T cell priming in DT treated CCR2-DTR mice. CCR2-DTR mice were infected and treated with DT or PBS on days 7, 9 and 11 and received highly purified CD45.1+ IMs on days 8 and 10. (**A**) Mice were euthanized on day 12 post infection and flow cytometry was performed on lungs and mLNs to track the engraftment of adoptively transferred IMs. (**B**) The expression of cell surface markers CD11b, MHC II, CDIIc and CD103 by adoptively transferred IMs in the lung and mLN was
*Figure 3. Continued on next page*

*Figure 3. Continued*

determined. (**C**) CFU plots from day 12 mLNs of CCR2-DTR mice rescued with IMs. Dots marked in red represent mice that received double sorted IMs of greater than 99% purity. (**D**) CCR2-DTR mice received a dose of naive ESAT-6-specific C7 T cells the day before infection and were treated with DT on days 7, 9 and 11 and received purified IMs or PBS on days 8 and 10. Lungs were harvested on day 15 and ESAT-6-specific C7 T cells were visualized. (**E**) Cumulative data of experiment shown in (**D**) showing the total number of ESAT-6-specific C7 T cells in day 15 lungs. (**F**) CCR2-DTR mice were depleted on days 7, 9 and 11 and received CCR2 WT or CCR2 KO IMs on days 8 and 10. Day 12 mLNs were harvested for CFU counts. Each dot represents an individual mouse. Error bars denote SEM. Data are representative of two independent experiments.

The following figure supplements are available for figure 3:

**Figure supplement 1**. Sorting of IMs from CCR2-GFP mice. FLT3-, cKit-, GFP+ cells were sorted from the bone marrow of CCR2-GFP mice.

Although IMs acquired characteristics of dendritic cells and are essential for Mtb transport from infected lungs to mLNs, it remained unclear whether IMs directly primed Mtb-specific CD4 T cells. Therefore, we transferred C7 T cells into MHC class II-deficient CCR2-DTR mice, depleted monocytes 7–11 days following infection and adoptively transferred MHC class II-expressing purified monocytes on days 8 and 10 following infection. In contrast to our previous experiments with MHC class II-expressing recipients, C7 T cells were not primed in MHC class II-deficient recipients of MHC class II-expressing monocytes (*Figure 4A*). Adoptively transferred IMs trafficked to mLNs (*Figure 4—figure supplement 1*) and, as demonstrated in the previous experiments, corrected Mtb delivery to draining lymph nodes to levels seen in mice that had not been monocyte-depleted (*Figure 4B*). Thus, defective antigen transport to the draining mLN could not explain the lack of CD4 T cell proliferation. In a reciprocal experiment, we transferred MHC class II-deficient IMs into Mtb-infected, MHC class II-expressing CCR2-DTR mice on the same schedule described above and quantified the C7 T cell response. *Figure 4C* demonstrates that the magnitude of the C7 T cell response was similar in monocyte-depleted mice rescued with either MHC class II-expressing or deficient monocytes. Taken together, these experiments indicate that IMs primarily serve as carriers of Mtb, delivering live bacteria to mLNs and enabling but not directly priming CD4 T cells.

Classical DCs in lymph nodes and their progenitors express the cDC-specific transcription factor zbtb46, whereas IMs do not. To determine whether classical DCs are required for Mtb-specific CD4 T cell priming, we used zDC-DTR mice, in which DT administration leads to a loss of classical DC populations (*Meredith et al., 2012a*; *Meredith et al., 2012b*). We transferred C7 T cells into zDC-DTR mice, infected the mice with Mtb and administered DT on days 7–11 following infection. Depletion of classical DCs resulted in markedly reduced C7 T cell expansion (*Figure 4D*, *Figure 4—figure supplement 2*) but did not reduce the number of live Mtb in mLNs (*Figure 4E*). FLT3L-deficient mice, which lack classical DCs but not IMs, also have markedly reduced CD4 T cell responses following Mtb infection (*Figure 4F*) despite the presence of Mtb in the mLN (*Figure 4—figure supplement 3*). These results indicate that classical DCs, while dispensable for trafficking of live Mtb from infected lungs to mLNs, are essential for CD4 T cell priming.

These experiments have refined our understanding of the role of IMs during the initiation of adaptive immune defense against *M. tuberculosis* infection. During the second week of murine infection, IMs play the critical role of delivering live Mtb to draining mLNs, an indispensable step for Mtb-specific CD4 T cell priming. Our results extend previous reports that IMs influence CD4 T cell priming (*Leon et al., 2007*; *Hohl et al., 2009*; *Nakano et al., 2009*; *Rivera et al., 2011*) by demonstrating that IMs serve as transporters of live bacteria from the site of infection to the site of T cell priming.

In contrast to reports from other disease models demonstrating that IMs differentiate into DCs and then directly prime CD4 T cells (*Cheong et al., 2010*; *Zigmond et al., 2012*) our adoptive transfer studies using MHC class II KO mice reveal that IMs, despite acquiring characteristics of DCs, do not prime Mtb-specific CD4 T cells in vivo. Why IMs fail to prime DCs in this setting remains unclear. Evidence that moDCs generated in vitro can stimulate T cell responses suggests that IMs can process and present antigen (*Schreurs et al., 1999*). Recovery of several hundred adoptively transferred IMs from the mLNs of infected mice also suggests that sufficient numbers of IM were present to prime T cells. One possible explanation is that IMs, by virtue of being infected with Mtb, are ineffective at stimulating Mtb-specific lymphocytes, in contrast to resident dendritic cells that have not been infected (*Wolf et al., 2007*).

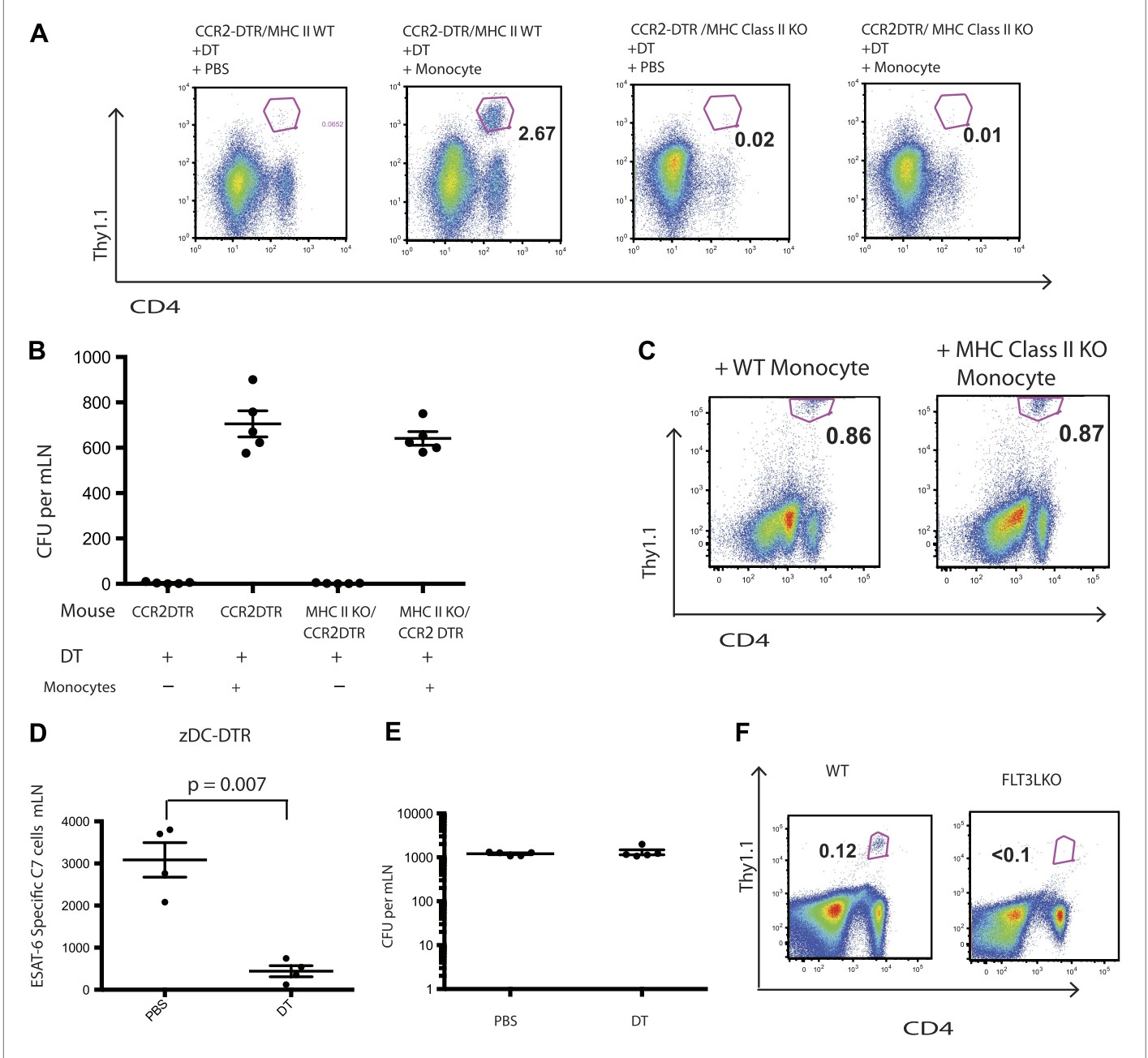

**Figure 4**. IMs do not prime CD4 T cells directly. CCR2-DTR mice that were either MHC class II KO or WT received naive ESAT-6-specific C7 T cells the day before infection and were treated with DT or PBS on days 7, 9 and 11 and received purified IMs on days 8 and 10. (**A**) ESAT-6-specific C7 T cells in lungs harvested on day 15. (**B**) Mtb CFUs from mLNs of mice harvested on day 12. (**C**) CCR2-DTR mice received naive ESAT-6-specific C7 T cells the day before infection and were treated with DT or PBS on days 7, 9 and 11 and received either MHCII WT or MHCII KO IMs on days 8 and 10. Flow cytometry was performed on lungs harvested on day 15. (**D**) zDC-DTR mice received naive ESAT-6-specific C7 T cells the day before infection and cDCs were depleted on day 7, 9 and 11, and the number of ESAT-6-specific C7 T cells was determined in mLNs harvested on day 12. (**E**) The number of CFU in mLNs of PBS and DT treated zDC-DTR mice. (**F**) FLT3LKO mice received naive ESAT-6-specific C7 T cells the day before infection and lungs were harvested on day 15 to quantify the ESAT-6-specific C7 T cells. Each dot represents an individual mouse. Error bars denote SEM. Data are representative of two independent experiments.

The following figure supplements are available for figure 4:

**Figure supplement 1**. Adoptively transferred IMs traffick to the mLNs of MHC Class II KO mice.

*Figure 4. Continued on next page*

*Figure 4. Continued*

**Figure supplement 2**. Priming of ESAT-6-specific C7 T cells is reduced in zDC-DTR mice.

**Figure supplement 3**. Mtb traffick to the mLN of FLT3LKO mice is unimpaired.

Antigen transport and T cell priming are often considered the two major functions of DCs. A number of studies, however, have demonstrated that multiple DC subsets can be involved in priming of naive T cells (*Itano et al., 2003*). Other studies characterizing CD8 T cell responses to cutaneous viral infection have demonstrated a role for CD103+ DCs in transport and priming with potential contributions by inflammatory monocytes (*Bedoui et al., 2009*; *Edismo et al., 2009*). One previous study has suggested that IMs carry fungal antigens from a site of cutaneous inoculation to draining LNs and transfer antigens to LN resident DCs (*Ersland et al., 2010*). Our study extends this finding to live pulmonary infection with Mtb. IMs transport bacteria to lymph nodes and transfer antigen to classical dendritic cells prior to CD4 T cell priming.

The mechanism of antigen transfer in mLNs remains undefined. It is possible that cross-dressing (*Wakim et al., 2011*), in which infected cells pass MHC molecules to uninfected cells, plays a role during Mtb infection. We did not detect transfer of MHC class II molecules from host cells to adoptively transferred, MHC class II-deficient monocytes. The number of MHC/peptide complexes required per APC to stimulate T cell responses, however, may be very small and below our level of detection. That said, given our finding that MHC class II-deficient IMs effectively complemented monocyte-depleted, MHC class II sufficient mice while MHC class II-expressing monocytes could not correct deficient T cell priming in MHC class II-deficient, monocyte-depleted mice, we believe that transfer of MHC class II molecules in either direction plays a minimal role in T cell priming during Mtb infection. Future studies will determine whether live Mtb are transferred to classical DCs, or whether transfer principally involves the movement of processed or unprocessed proteins from infected monocytes to uninfected DCs.

## Materials and methods

### Mice

C57BL/6 and MHC class II-deficient mice were purchased from the Jackson Laboratory. The generation of ESAT-6-specific C7 TCR transgenic, CCR2-DTR, CCR2-GFP and zDC-DTR mice were previously described (*Wolf et al., 2007*; *Gallegos et al., 2008*; *Hohl et al., 2009*). For depletion experiments, mice were injected i.p. with 20 ng/g body weight DT. All mice were bred and maintained under specific pathogen-free conditions at the Memorial Sloan Kettering Research Animal Resource Center. Sex-and age-matched controls were used in all experiments according to institutional guidelines for animal care. All animal procedures were approved by the Institutional Animal Care and Use Committee of the Memorial Sloan-Kettering Cancer Center.

### Aerosol infections with *M. tuberculosis*

*M. tuberculosis* Erdman was grown in 7H9 media, and log phase cultures were diluted to $8 \times 10^6$ bacilli per millimeter and sonicated before infection with an aerosol exposure system (Glass-Col). The volume of suspension, and exposure time were calibrated to deliver ~100 CFU per animal. To determine infection dose, three mice were killed 1 day after infection and lungs were homogenized in PBS/0.05% Tween-80, and half the lung homogenate was plated. At various intervals after infection the left lung was harvested from individual mice and homogenized in PBS/0.05% Tween-80. Serial dilutions were made in PBS/0.05% Tween-80 and plated onto Middlebrook 7H10 agar (BD Biosciences). After 3 weeks of incubation at 37°C in a 5% $CO_2$ atmosphere, colonies were counted.

### Adoptive IM transfer

Inflammatory monocytes were harvested from the bone marrow of CCR2-GFP mice. CD4 T cells and NK+ cells were removed using antibody depletion kits from Miltenyi Biotec. The remaining cells were then stained with anti FLT3 PE and anti C-kit PE antibodies and EGFP+, PE- cells were sorted by the Memorial Sloan-Kettering Cancer Center flow cytometry core facility. The cells were re-suspended in PBS and $2 \times 10$ (*Kipnis et al., 2003*) cells per mouse were injected via the tail vein.

## Statistics

All data are presented as the arithmetic mean ± SEM. Statistical validation was done with the Student's *t* test. $p < 0.05$ were considered significant, $p > 0.05$ were considered insignificant.

## Acknowledgements

We thank L Lipuma, H Yan and S Reddy for technical support, MC Nussenzweig for providing the zDC-DTR mice, and J van Heijst, R Samstein and J Schneider for discussions.

## Additional information

### Funding

| Funder | Grant reference number | Author |
| --- | --- | --- |
| National Institute of Allergy and Infectious Diseases | AI080619-01 | Miriam Samstein, Ingrid M Leiner, Bože Sušac, Michael S Glickman, Eric G Pamer |
| National Institute of Allergy and Infectious Diseases | AI039031 | Miriam Samstein, Ingrid M Leiner, Bože Sušac, Michael S Glickman, Eric G Pamer |
| National Institute of General Medical Sciences | T32GM07739 | Miriam Samstein |

The funders had no role in study design, data collection and interpretation, or the decision to submit the work for publication.

### Author contributions

MS, Conception and design, Acquisition of data, Analysis and interpretation of data, Drafting or revising the article; HAS, Provided zDC-DTR mice and helped design related experiments; IML, BS, Maintained Clone7, CCR2-DTR and CCR2-GFP mice, generated MHC Class II KO CCR2-GFP and CCR2-DTR mice, and planned related experiments; MSG, Gave conceptual advice, supervised experiments, and edited the manuscript; EGP, Gave conceptual advice, supervised experiments and data analysis, and edited the manuscript

### Ethics

Animal experimentation: This study was performed in strict accordance with the recommendations in the Guide for the Care and Use of Laboratory Animals of the National Institutes of Health. All of the animals were handled according to protocols approved by the Institutional Animal Care and Use Committee (IACUC) of the Memorial Sloan-Kettering Cancer Center (protocols 01-11-030 and 00-05-066).

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
