## [Decision Letter]

Thank you for sending your work entitled “Essential yet limited role for CCR2^+^ inflammatory monocytes during Mycobacterium tuberculosis-specific T cell priming” for consideration at *eLife*. Your article has been favorably evaluated by a Senior editor and 2 reviewers, one of whom is a member of our Board of Reviewing Editors.

The Reviewing editor and the other reviewer discussed their comments before we reached this decision, and the Reviewing editor has assembled the following comments to help you prepare a revised submission.

This is an elegant and straightforward study that makes an important contribution on the role of different APCs in transport and presentation of antigen in vivo in a biologically relevant system.

The only open aspect is the mechanism of antigen transfer from inflammatory monocytes to resident DC. It would be interesting if the authors could provide insights on why the monocytes that carry Mtb to lymph nodes do not present antigen to CD4^+^ T cells. Is there a problem with their localization within lymph nodes? Or are they incompetent for antigen processing and presentation (their antigen presentation capacity could be tested ex vivo)? Or are they too few to function as APC and their role is limited to spreading a live organism? In addition the authors may also consider the alternative possibility that resident DCs transfer their MHC class II molecules to inflammatory monocytes (a process related to “cross dressing”). In this case DC depletion would block T cell activation because the source of MHC class II is absent. This possibility could be addressed by asking whether there is any MHC class II on class II KO inflammatory monocytes after adoptive transfer and infection. While these experiments would provide a mechanistic explanation to the results, they are not essential to support the major conclusions of the study and consequently the authors may address these issues in the Discussion.

---

## [Author Response]

We thank the editor and reviewers for the review of our manuscript. We have addressed the major issues raised by the review with additional experiments and we have modified the Discussion of our manuscript. As pointed out, the mechanism of antigen transfer between monocytes and resident DCs remains unclear. Many in vitro studies have demonstrated the ability of monocyte-derived DCs to prime T cells, suggesting that IM-derived cells are not intrinsically incapable of antigen processing, presentation or T cell priming. Our ability to detect several hundred adoptively transferred IMs per mLN by flow cytometry suggests that a paucity of these cells is an unlikely explanation for their inability to prime T cells. We believe, and there is published evidence for this from other laboratories, that live Mtb infection interferes with MHC class II antigen processing and presentation. We have inserted statements in the Discussion to address these interesting possible explanations for our findings.

The reviewers also raised the possibility that our findings of CD4 T cell priming following transfer of MHC class II-deficient IMs might be explained by transfer of MHC molecules from mLN resident cells to IMs. Because this phenomenon, referred to as cross-dressing, might contribute to CD4 T cell priming during Mtb infection, we performed a repeat experiment in an attempt to detect MHC II molecules on MHC II-deficient IMs following their transfer into Mtb infected, MHC-class-expressing mice. Although we did not find evidence for cross-dressing using this approach, since it is possible that very few MHC molecules are required per cell for T cell priming, we are unable to rule out its contribution. That said, given our finding that MHC class II-deficient IMs effectively complemented monocyte-depleted, MHC class II sufficient mice while MHC class II-expressing monocytes could not correct deficient T cell priming in MHC class II-deficient, monocyte depleted mice, we believe that transfer of MHC class II molecules in either direction plays a minimal role in T cell priming during Mtb infection.

The manuscript has been corrected to reflect both the number of mice per experiment as well as the number of repetitions that were performed of each experiment.